# Prospects and Current Challenges of Extracellular Vesicle-Based Biomarkers in Cancer

**DOI:** 10.3390/biology13090694

**Published:** 2024-09-04

**Authors:** Samuel R. Lawrence, Karan M. Shah

**Affiliations:** Division of Clinical Medicine, School of Medicine & Population Health, The University of Sheffield, Beech Hill Road, Sheffield S10 2RX, UK

**Keywords:** extracellular vesicles, biomarker, liquid biopsy, cancer diagnosis

## Abstract

**Simple Summary:**

Cancer presents a significant global health challenge, particularly in ageing populations, emphasising the importance of early detection to facilitate personalised care and optimised treatment to improve patient outcomes. Liquid biopsy offers a minimally invasive approach for isolating and profiling circulating tumour-derived components with potential for cancer diagnosis and monitoring. Tumour-derived extracellular vesicles are secreted from cancer cells and harbour cancer-specific cargo with enhanced stability within the blood, indicating their potential utility as a diagnostic biomarker. However, translating extracellular vesicle biomarkers into clinical practice faces significant hurdles. This critical review evaluates the current landscape, challenges, and future directions of tumour-derived extracellular vesicle biomarkers, proposing strategies to overcome translational barriers. The successful integration of a liquid biopsy-based assessment of tumour-derived extracellular vesicles could transform cancer diagnostics and management.

**Abstract:**

Cancer continues to impose a substantial global health burden, particularly among the elderly, where the ongoing global demographic shift towards an ageing population underscores the growing need for early cancer detection. This is essential for enabling personalised cancer care and optimised treatment throughout the disease course to effectively mitigate the increasing societal impact of cancer. Liquid biopsy has emerged as a promising strategy for cancer diagnosis and treatment monitoring, offering a minimally invasive method for the isolation and molecular profiling of circulating tumour-derived components. The expansion of the liquid biopsy approach to include the detection of tumour-derived extracellular vesicles (tdEVs) holds significant therapeutic opportunity. Evidence suggests that tdEVs carry cargo reflecting the contents of their cell-of-origin and are abundant within the blood, exhibiting superior stability compared to non-encapsulated tumour-derived material, such as circulating tumour nucleic acids and proteins. However, despite theoretical promise, several obstacles hinder the translation of extracellular vesicle-based cancer biomarkers into clinical practice. This critical review assesses the current prospects and challenges facing the adoption of tdEV biomarkers in clinical practice, offering insights into future directions and proposing strategies to overcome translational barriers. By addressing these issues, EV-based liquid biopsy approaches could revolutionise cancer diagnostics and management.

## 1. Introduction

Despite recent technological and therapeutic advancements, cancer remains a significant and extensive public health challenge. In approximately 60% of countries worldwide, cancer is the first or second leading cause of mortality, underscoring a critical need for healthcare advancements to improve patient outcomes [1]. To this aim, the utilisation of cancer biomarkers in clinical practice offer specific and measurable indications of cancer stage classification, patient risk/prognosis, and identification of residual/relapsing disease [2]. Early cancer identification facilitates the adoption of personalised medicine strategies, thereby improving individual prognoses and cancer survival rates and ultimately alleviating the strain on healthcare providers. Importantly, the identification and utilisation of novel cancer biomarkers as clinical tools to advance cancer diagnosis and treatment aligns with the objectives delineated in the United Kingdom’s National Health Service ‘Long Term Plan’ (January 2019), aiming to improve patient five-year survival outcomes [3]. A notable example of an approved and widely used cancer biomarker in clinic is the prostate-specific antigen (PSA) test. This assay facilitates the early detection of prostate cancer and improves the clinical management of the disease, despite controversies surrounding false positive results and cancer overdiagnosis [4]. Notably, a constraint of the PSA test is its indiscrimination between patients at risk of cancer-related premature mortality, and those whose disease may remain indolent without necessitating surgical or therapeutic intervention [5]. Biomarkers that offer diagnostic and prognostic insights are highly sought after in clinical practice, with the important characteristics of a clinically valuable biomarker detailed in Figure 1 [6].

An extracellular vesicle (EV), as defined by the International Society for Extracellular Vesicles (ISEV), is a generic term used to describe particles enclosed within a lipid bilayer, that are released from cells, and which cannot replicate [7]. Identifying EVs from specific biogenesis pathways is difficult; however, within the literature, common subtypes referred to include the endosome-derived ‘exosomes’ or the plasma membrane-origin ‘ectosomes’ (microvesicles) (Figure 2). Despite ISEV recommendations to use operational terms for EV nomenclature based on physical characteristics, biochemical composition, and cell of origin descriptions, the current literature remains burdened with manifold definitions. Therefore, for the purpose of this review, any denomination of EV encountered will be referred to thereafter as either tumour-derived EV (tdEV) or EV. The interest and relevance of EVs within the field of cancer biomarkers stems from emerging evidence suggesting their involvement in cancer progression via intercellular communication, suppression of immune responses, and inducing metastasis malignancy-associated phenotypes [8,9,10]. EVs are known to harbour tumour-derived cargo including RNA/DNA, proteins, and lipids that are distinct from the background host EV population [11]. For instance, tdEVs have been found to carry DNA that reflects the mutational landscape of their cells of origin. Within the context of melanoma, one study isolated EVs from six human melanoma metastatic tissues and compared the DNA mutations with both tumour tissue DNA and plasma DNA. Using ultra-sensitive sequencing (SiMSen-seq), researchers investigated a panel of 34 melanoma-related genes. Notably, mutations were detected in six genes (BRAF, NRAS, CDKN2A, STK19, and PPP6C) in the EVs isolated from melanoma tissues, closely mirroring the mutational profile of the originating tumours [12]. It is known that, in cancer development, hypermethylation of the DNA is an early event, and it remains methylated as cancers progress [13]. Indeed, methylation and other epigenetic statuses can also be exploited as a diagnostic biomarker [14,15]. In gastric cancers, the methylation status of BarH-like homeobox protein (*BARHL2*) from gastric juice-derived EVs has been shown to have 90% sensitivity and 100% specificity, with respect to discriminating patients from controls [16]. More recently, Maire et al. (2021) demonstrated that the genome-wide methylation profile and copy number variations (CNVs) of glioblastoma-derived EVs mirror the landscape observed in the original tumours [17].

Histone acetylation has also shown to play an important role in tumorigenesis, and acetylome analyses of tdEVs from breast cancer cells have found a distinct acetylated protein profile that may mirror the cell of origin and thus serve as a good biomarker candidate [18]. Separately, Al-Nedawi et al. (2008) observed that modified U373 glioblastoma cells harbouring a mutated epidermal growth factor receptor (EGFR) secreted EVs containing the mutated receptor, contrasting EVs derived from parental U373 cells [19]. Moreover, upon exposure of parental cells to EVs carrying mutated EGFR, the authors observed an increased cellular expression of vascular endothelial growth factor. This is indicative of cell transformation suggesting EV-mediated cellular communication is involved in the promotion of malignant processes and carcinogenesis. Identifying an association between EVs and a clinical condition or biological phenotype corresponds to the first characteristic of an effective biomarker (Figure 1), suggesting the potential clinical utility of EVs, pending the development of reliable detection methodologies.

Recent research indicates an emerging interest in using EVs within cancer diagnostics, evidenced by a study conducting long RNA sequencing from serum-based EVs obtained from early-stage lung adenocarcinoma patients. This study successfully identified a 23-gene signature capable of distinguishing patients from benign controls with high analytical sensitivity, specificity, and accuracy (93.75%, 85.71%, and 88.24%, respectively) [20]. For lung cancer alone, the significance of this discovery lies in the potential integration of EV sequencing analysis with the current diagnostic standard of low-dose computed tomography (CT). Considering indications from the U.S. National Lung Screening Trial reporting that 96.4% of low-dose CT positive screening results are false positives, a combinatorial approach involving the analysis of EV cargo may help mitigate this and even improve patient disease stratification [21]. Separately, Mercy BioAnalytics recently received USD 41 million in Series A financing to optimise their assay analysing co-localised proteins on the surface of single EVs to improve early cancer detection [22]. This indicates potential for not only their technology, but the field as a whole, by demonstrating the increasing recognition of EVs as a potential tool for cancer diagnostics. It is based on such indications that this critical review aims to discuss and broadly evaluate the prospects, challenges, and future work facing EVs regarding their translation into clinical practice, to shed light on this promising area of cancer diagnostics.

## 2. Prospects

Within the context of cancer research and analysis, liquid biopsy has generated great interest through its ability to identify various tumour-derived components, including circulating tumour cells (CTCs), cell-free DNA (cfDNA), and EVs, in the bodily fluids of cancer patients [23]. Liquid biopsy employs a technically simple, cost-effective, and minimally invasive repeatable approach to sample collection [24]. This is essential for providing patient prognostic indications, both throughout a treatment course and for research-oriented functions. The subsequent molecular profiling of samples through genomic and proteomic analysis of isolated components has the potential to improve cancer detection at earlier stages, guide personalised medicine strategies, and aid in therapy response monitoring. Moreover, unlike traditional tissue biopsies, liquid biopsy approaches are not limited by tumour accessibility and reflect the systemic characteristics of all bodily tissue. Consequently, they offer a comprehensive depiction of the heterogenous nature of cancer which is important for developing a clinically valuable diagnostic assay [25].

### EVs as a Promising Approach to Cancer Detection

The burgeoning body of evidence highlighting the involvement of EVs in cancer pathogenesis suggests a promising avenue for innovative early cancer detection strategies [26,27,28]. As a result, there is growing interest in the utilisation of tdEVs as both cancer biomarkers and therapeutic targets. Importantly, EVs provide access to patient physiological data in real time from various biofluids including urine, blood, stool, and saliva. This feature presents a distinct advantage over the traditional tissue biopsy approach for cancer diagnosis and monitoring. Previously, liquid biopsy has concentrated solely on the analysis of CTCs and tumour-originating cfDNA, with several diagnostic assays being commercially available, including the approved FoundationOne^®^ Liquid CDx test [29]. The extension of this approach to include the analysis of tumour-derived EVs (tdEVs) may provide patient benefits, pending the development of reliable and robust diagnostic platforms. To date, few studies have accurately quantified and compared the presence of tdEVs and CTCs within patient samples. However, available lines of evidence suggest that tdEVs are notably more abundant in blood compared to CTCs, with cargo including RNAs demonstrating acceptable half-lives, signifying the potential utility of tdEVs as valid cancer biomarkers [22,25,30,31].

Supporting this, Nanou et al. (2020) identified significantly higher tdEV counts compared to respective CTC counts by at least an order of magnitude in castration-resistant prostate cancer, metastatic breast cancer, metastatic colorectal cancer, and non-small cell lung cancer [32]. The robustness of this study’s methodology is attributed to the large patient cohort sizes and comprehensive diversity in cancer types assessed. Conversely, another study revealed the absence of tdEVs in prostate cancer patients, despite the presence of detected CTCs [33]. The discrepancy observed between these studies may arise from the authors’ use of differentially gated ACCEPT enumeration software for the quantification of tdEV populations, thereby imposing limitations on the interpretations of their findings.

To improve data integrity in the future, researchers may consider using an EV-specific detection method, such as transmission electron microscopy, in conjunction with computer-assisted image analysis, such as ExosomeAnalyzer software [34]. Additionally, the authors of both papers indicate that the EV isolation method utilised in their research enriched only large EVs, constituting an estimated <1% of total tdEVs present within blood serum, preventing the holistic analysis of the tdEV landscape within samples [32]. Despite experimental shortcomings, current evidence suggests that tdEVs are present in significantly higher numbers than total CTCs, theoretically supporting the prospect of utilising EVs as cancer biomarkers. However, quantifying and analysing tdEVs may primarily serve to infer cancer stage and patient prognoses, while the analysis of CTCs might prove preferential to indicate cancer diagnosis while being technologically simpler to isolate [35].

Importantly, there remains a necessity in the field to optimise the enrichment of smaller tdEVs, in order to achieve a comprehensive understanding of the complete analytical potential of tdEVs. A prospective detection strategy for the future of EV-based diagnostics is utilisation of an optimised dynamic light scattering (DLS) approach for EV analysis. Kogej et al. (2021) supports this, demonstrating the ability of DLS to characterise extracellular nanoparticles in the bodily fluids of benign and malignant ovarian cancer patients, enabling patient stratification [30]. However, due to the heterogenous nature of EVs, reaching a defined tdEV biomarker threshold using this approach might require large sample volumes, which are likely to be impractical in a clinical environment. Nonetheless, a DLS approach utilising laser beams and the detection of resulting photon scatter to analyse EV particle size is highly sensitive, with a large defined detectable size range of 1–6000 nM [36]. This technique is shown to yield highly accurate particle size distribution data while permitting fast sample processing/measurement times [37]. This is significantly advantageous when compared to current EV characterisation techniques involving ultracentrifugation or size-exclusion chromatography. Furthermore, DLS harbours the potential to analyse complex biological fluid, thereby removing the need for prior sample dilution/fractionation, theoretically enabling bulk and reduced-cost analyses of samples [38].

However, there is contention within the field regarding the reported reduced accuracy of DLS size measurements for small tdEVs within polydispersed solutions [36,39], perhaps indicating a need in the future to integrate DLS diagnostic solutions with other next-generation technologies, such as atomic force microscopy (AFM), in a combinatorial approach to EV analysis. This type of nanotechnology is capable of atomic scale imaging operating through a cantilever equipped with a nanoscale tip, which can scan the surface of an EV using a defined tip force while a laser beam reflected onto a photodiode precisely monitors the deflection of the cantilever, thus generating topographic images of the EV’s surface [40]. An advantage of using AFM for the biophysical characterisation of EVs can be attributed to the technology’s capability to deposit analyte on a solid substrate and image samples either dried or within cell culture media, with no need for staining or fixation providing a degree of flexibility within the workflow [41]. The ability to analyse and generate topographic images of single membrane proteins at a lateral and vertical resolution of <1 nm and <0.1 nm, respectively, while also generating insights into specific vesicular mechanical characteristics, suggests clinical potential for the identification of tdEV nanomechanical fingerprints. One study using AFM to investigate the mechanical properties of human breast cancer-derived tdEVs identified that vesicle stiffness and osmotic pressure increased with increasing disease malignancy, indicating that this technology may also aid in patient stratification into low- and high-risk groups [42]. AFM’s utility is, however, limited to surface analysis and low-throughput particle engagement, and can only analyse several hundred nanosized particles within an hour [43]. The workflow still requires significant sample processing and is limited by the bottleneck of pure EV enrichment, suggesting that AFM can only aid in the identification of specific tdEVs and provide an indication of contaminant levels when used in conjunction with other diagnostic techniques.

Crucially, the current literature indicates that EVs and its contents exhibit long-term stability and degradation resilience during freezing, storage, and thawing processes [44]. This is important to consider, suggesting potential flexibility regarding sample processing and handling, with minimal sample degradation during transport from clinic to diagnostic laboratories. From a diagnostics and research perspective, this is essential for the continued discovery/validation of novel tdEV biomarkers using clinical samples, or for treatment monitoring purposes where patient sample biobanking may occur over an extended period. Further work in this area would prove beneficial, as the latest ISEV expert position paper (2023) did not reach an agreement indicating optimal EV storage processes for maintaining sample integrity [45,46]. Interestingly, existing accounts in the literature attempting to address this issue are divided, with numerous studies reporting that freeze–thaw cycles do not cause significant changes in EV count [47,48,49,50]. For example, Jayachandran et al. (2012) observed no significant loss of EVs when stored at either −40 °C or −80 °C for up to one year, including up to three freeze–thaw cycles [50]. Contradictorily, other investigations suggest significant alterations in EV physical characteristics after only six months of storage, claiming that −80 °C storage minimised but did not prevent EV loss [51,52]. To date, the significance and impact of varying sample storage conditions on the effective analysis and diagnostic potential of EVs remains unexplored. Extensive diversity regarding the biological source and composition of EVs might be responsible for observed differences between these investigations. For instance, certain EV subpopulations with reduced lipid bilayer stability may have heightened sensitivity to temperature fluctuations and require more stringent storage conditions than others [53]. Overall, the current evidence suggests that EV integrity may be preserved at −80 °C for up to one year, with some resistance to freeze–thaw cycling. This likely presents a notable clinical benefit in utilising EVs as biomarkers, indicating their inherent sample storage flexibility, enabling reliable biomarker analysis, and facilitating the longitudinal monitoring of patient disease and treatment response over time.

## 3. Challenges

An ideal method for EV isolation should be inexpensive, reasonably fast, and allow for enrichment from several biofluid sources. In order for EVs to be considered as validated cancer biomarkers, they must undergo analytical validation, clinical validation, and clinical utility assessment, as required by international regulatory agencies, to demonstrate appropriate clinical benefit [54]. Passing analytical validation involves the provision of sufficient evidence suggesting that biomarker assay results are highly accurate and repeatable, while being both sensitive and specific [55]. Success at this stage would suitably demonstrate an association between EV type, quantity, or contents and a specific cancer diagnosis or prognosis. Becoming clinically validated involves illustrating the biomarker’s utility by demonstrating an acceptable assay performance within clinical trials that will beneficially influence clinical treatment decision-making [6]. Crucially, there are several obstacles preventing the translation of the tdEV biomarker into clinical practice, including extensive EV population heterogeneity, the presence of contamination during EV enrichment, and the lack of method standardisation (see Table 1).

### 3.1. EV Heterogeneity

EVs are known to be an incredibly heterogenous and diverse set of particles generated from differing biogenesis pathways with stratified EV populations based on their size, density, biochemical composition, and cell-of-origin. The traditional classification of EVs condenses the wide range of particle types into three categories: exosomes, which are formed through the exocytosis of multivesicular endosomes (30–150 nm diameter, 1.13–1.19 g/mL); microvesicles, which are generated by the outward blebbing of the plasma membrane (100–1000 nm diameter, 1.08–1.19 g/mL); and apoptotic bodies, which are remnants of the late stages of apoptosis (50–5000 nm diameter, 1.12–1.23 g/mL) (Figure 2) [36,56,57]. Notably, the biochemical composition of EVs is highly diverse and reflects both common EV markers and cell-specific components (see Table 2). This ultimately poses a challenge in optimising tdEV quantification and characterisation methods for effective biomarker analysis, where identifying tdEVs from the host background of isolated EVs remains difficult [7]. Extensive EV physical heterogeneity makes establishing tdEV-specific markers suitable for diagnostic applications challenging, ultimately preventing effective biomarker analysis and hindering their adoption into clinical practice [58]. For instance, classical exosomes can be identified by the tetraspanin markers CD9, CD63, and CD81, in contrast to non-classical exosomes, which lack these markers. Classical microvesicles and large oncosomes can be identified through Annexin A1 and adenosine diphosphate ribose (ADP) ribosylation factor 6, while small microvesicles uniquely express TSG101 and ARRDC1, despite all originating through the plasma membrane shedding pathway [59]. One study investigating EV-mediated communication within glioma identified that proneural stem cells release EVs largely devoid of surface markers, compared to those originating from mesenchymal stem cells.

**Table 1 biology-13-00694-t001:** Summary of the main challenges and potential solutions for the clinical translation of tdEVs as biomarkers.

Challenges	Indications	Potential Solutions
Extensive EV physical heterogeneity (size, density, and composition) makes tdEV quantification difficult.	- Within glioma, proneural stem cells release EVs largely devoid of markers, while those derived from mesenchymal stem cells uniquely express CD9, CD63, and CD81, indicating intra-disease heterogeneity [60]. - Up to 5000 distinct protein signals have been detected in the EV-associated proteome of a typical cancer cell population [61,62,63,64,65,66,67].	- Identify and collate appropriate reference genes for the analysis of EV populations derived from multiple tissues and cell types. - Utilise highly sensitive gene amplification technologies (such as RT-qPCR) for accurate EV nucleic acid quantification, targeting known gene variants. - Identification of tdEV-specific proteins with subsequent proteomic profiling, using highly specific antibody-based approaches.
Enriching EVs often results in the co-isolation of contaminating proteins, which interfere with downstream analysis.	- Maintaining cellular integrity during EV enrichment to prevent intracellular debris release is a technical challenge. - The commonly used method of size exclusion chromatography (SEC) frequently results in the co-isolation of contaminating lipoproteins [68].	- Employ additional isolation methods such as density gradient separation. - Optimise immunocapture techniques combined with light scattering flow cytometry for high-purity EV isolation.
The lack of standardised enrichment methodologies leads to excessive variation and inconsistencies in tdEV detection.	- Variations in pore sizes for SEC, or in relative centrifugal force for density gradient separation, will enrich different sub-fractions of tdEVs [69].	- Develop a universal approach for enriching different EV types, shifting to standardised, scalable, and accessible technologies and facilitating cost-effective scale-up opportunities.

EVs expressing the classical exosome tetraspanins CD9, CD63, and CD81 evidence EV heterogeneity even within the same disease [60]. Other groups have also reported extensive EV cargo heterogeneity, suggesting that the average EV-associated proteome of a uniform cultured cancer cell population can contain up to 5000 distinct protein signals [61,62,63,64,65,66,67]. The inherent variation associated with EV characteristics presents challenges for researchers in identifying a specific gene or protein signature commonly expressed within a population of tdEVs, thereby hindering their diagnostic utility. Notably, the use of gene amplification technologies, such as reverse transcription quantitative PCR, which are highly sensitive and economically viable tools, enables the accurate quantification of EV nucleic acid cargo contents, opening up avenues for novel tdEV-based biomarker signature discovery [70]. However, due to widespread EV heterogeneity, the task of identifying a suitable and validated housekeeping gene to function as an experimental internal control remains a significant challenge in the field. It is important for researchers to identify reference genes appropriate for the analysis of EV populations derived from multiple tissues, in order to ensure the comprehensive and representative analysis of tdEV biomarker data in a clinical context [71,72].

Ultimately, this has resulted in inconsistent interpretations of EV gene expression profiles, thereby hindering the widespread adoption of molecular-based gene technologies for the standardised quantification of tdEV nucleic acid cargo [73]. The process of identifying and validating specific markers or signatures that ubiquitously define tdEV subtypes, originating from different diseases and encompassing the biological variation present between patients, constitutes a costly and time-consuming endeavour. However, one research initiative successfully performed the proteomic profiling of plasma-derived EVs from cancer patients and identified 51 and 19 tdEV-specific proteins unique to pancreatic adenocarcinoma and lung adenocarcinoma, respectively. This was completed using high-resolution/high-mass accuracy nano-liquid chromatography tandem mass spectrometry (MS) data, with the validation and quantitation of proteins performed experimentally using enzyme-linked immunosorbent assays (ELISAs). These findings suggest the prospective ability of tdEV proteomic analysis to discriminate between cancer and non-cancer patients, highlighting the importance of further investigation in this area [74]. Following EV enrichment, commonly used methods to interrogate the proteomic landscape include Western blotting, ELISAs, MS, and flow cytometry, which are powerful tools for detecting and quantifying tdEV-specific proteins. However, despite being highly sensitive techniques, they appear impractical for clinical use due to their labour-/resource-intensive nature [75].

**Table 2 biology-13-00694-t002:** EV types with associated markers.

EV Type	Marker	References
Exosome	CD9, CD63, CD81, TSG101, ALIX, HSP70, HSP90, HSP60, HSP27, HSPA8, Rab27a, Rab27b, syntenin-1, flotillins, ceramides, cholesterol, sphingomyelin, GPC1, CD147	[76,77]
Microvesicle	CD9, CD63, CD81, TSG101, ALIX, HSP70, HSP90, HSP60, HSP27, HSPA8, actin, myosin, ADP-ribosylation factor 6, annexin A1	[59,76,77,78]
Apoptotic body	CD9, CD63, CD81, TSG101, ALIX, HSP70, HSP90, HSP60, HSP27, HSPA8, histone H3, Caspase 3, Phosphatidylserine, annexin V	[59,76,77]
Cell Type Specific Markers	CD41 (platelets), CD235a (erythrocytes), EpCAM (epithelial cells), EGFR (cancer cells)	[79,80,81]

### 3.2. Contamination during Isolation

A separate limitation of utilising tdEVs as cancer biomarkers revolves around the co-isolation of contaminating particles (Table 3) when releasing EVs from the extracellular matrix during the technically difficult enrichment process [82,83,84]. It is key to prevent cellular disruption and subsequent contamination with intracellular debris, which can result in interference in downstream analyses. Serum starvation during in vitro cell culture research is a versatile tool for experimental manipulation, commonly employed for purposes involving cell cycle synchronisation, studying cellular metabolic stress responses by serving as a model for nutrient deprivation, and manipulating cell differentiation and gene expression programmes [85,86,87]. However, this is shown to also induce cell apoptotic processes, which release intracellular proteins, increasing their levels within blood plasma [88]. Ultimately, contaminating proteins (such as albumin) during EV isolation will lead to reduced sample purity with inaccurate tdEV quantification.

Particularly when using colourimetric assays or size-distribution detection methods, such as DLS or nanoparticle tracking analysis (NTA), which at present do not distinguish between EVs and other similarly sized particles such as lipoproteins, this reduces diagnostic assay specificity and the subsequent clinical utility of an EV-based biomarker by preventing accurate downstream biochemical and biophysical analyses, thereby hindering patient prognostic indications. Using antibodies to remove lipoproteins from blood plasma prior to EV analysis has demonstrated potential, with a study reporting a median reduction of particle concentration by 62% when removing ApoB-exposing lipoproteins, as measured by NTA [89]. However, this provides no clinical utility, considering samples had to be diluted up to 400 times to use an economically viable concentration of antibodies, indicating lack of scale-up capability and feasibility in clinical practice. The current literature indicates that performing alternative methods, such as SEC, to elute EVs also results in the co-isolation of contaminating chylomicrons and very-low-density lipoproteins, as they overlap in diameter [68]. Density gradient separation was able to remove soluble proteins from samples prior to SEC; however, the resultant added assay complexity and time similarly indicate its lack of suitability for clinical application. The existence of only one commercially available EV-based cancer biomarker supports the present lack of suitable EV enrichment methodologies. Namely, the ExoDx Prostate Intelliscore (EPI) test is the only one able to discriminate between high-grade, low-grade, and benign prostatic disease [90]. However, due to challenges detecting tdEVs, this assay sacrifices low assay specificity (34%) for high sensitivity (92%) through a low analyte detection threshold, ultimately increasing the number of false positives, as identified in a 499-patient cohort [90,91]. Separately, the presence of contaminating Tamm–Horsfall protein (THP) within urine samples results in the masking of low-abundance tdEV-associated proteins and interferes with glycosylation analysis [92]. Overall, this illustrates the difficulties of exploiting tdEVs as biomarkers and make it hard to justify their utility when compared to alternative non-EV-based diagnostic options (Table 4).

**Table 4 biology-13-00694-t004:** Examples of clinically validated biomarker tests.

Marker	Sample Type	Cancer Type	Value	Detection Methodology	Relevance
Phosphatidylinositol-4,5-biphosphate 3-kinase catalytic subunit alpha (PIK3CA)	Freshly frozen tissue biopsy	Breast	Sens. 100% Spec. 100%	Digital droplet PCR	Activating PIK3CA mutations occurs in 20–30% of all breast cancer cases. Specific mutations act as prognostic factors for relapse-free survival [93].
Circulating tumour cells	Blood plasma	Breast, prostate, colorectal	Sens. 85% Spec. 94.45%	Antibody	CTC enumeration can help assess therapeutic response and prognosis in metastatic cancers [94].
Circulating cfDNA	Blood plasma	Colorectal	Sens. 83.1% Spec. 90%	NGS	Colorectal cancer screening in individuals at average risk for the disease [95].
Exosomal RNA (SPDEF, PCA3, ERG)	Urine	Prostate	Sens. 92% Spec. 34% NPV: 91% PPV: 36%	RT-qPCR	Able to discriminate between high-grade, low-grade, and benign disease [90].
Faecal haemoglobin	Stool	Colorectal	Sens. 92.1% Spec. 85.8%	Antibody	Detect the degradation products of blood in faeces and can help identify patients requiring investigation with the highest priority [96].

Sens. (sensitivity), spec. (specificity), NPV (negative predictive value), PPV (positive predictive value).

### 3.3. Method Standardisation and Validation

A key component of a successful tdEV biomarker discovery programme relies upon the development of optimised enrichment methodologies suitable for different disease contexts and biofluid sources. For instance, liquid biopsy isolation of blood (plasma and serum) and urinary-based tdEVs serve as direct indicators of pathophysiological alterations within the urogenital system, presenting opportunities for the assessment of prostate cancer [97,98]. Ideally, employing a methodically universal approach for EV enrichment would facilitate the expedited and cost-effective scale-up of EV-based biomarker development with clinical applications, accelerating the availability of new therapeutics. However, the absence of standardised EV isolation protocols likely leads to inconsistencies and variation in the detection and analysis of vesicular subpopulations. This negatively affects the reliability and consistency of EV-based diagnostic assays, while presenting significant challenges for EV-derived biomarker discovery initiatives. Differences in analytical equipment, such as varying pore sizes for size exclusion chromatography, and in software for data processing and image analysis can introduce variability in the analysis and characterisation of tdEVs, ultimately hindering their analytical potential. Furthermore, there are significant discrepancies regarding various isolation methods’ capacity to effectively separate EVs from co-aggregated proteins and lipids, as identified through NTA [57]. These variations lead to marked differences in EV yields and contaminant levels, emphasising the critical role of method selection in EV isolation protocols for accurate research outcomes. Poor selection may negatively affect downstream genomic and proteomic landscape analyses compromising EVs’ accuracy and reliability as biomarkers. A global survey gathering 196 researcher responses investigating commonly used EV isolation/characterisation methodologies revealed ultracentrifugation to be the Gold Standard method, despite being labour-intensive and necessitating large amounts (20–100 mL) of starting sample volume [99]. This indicates potential challenges for clinical application by precluding the ability of clinicians to take regular liquid biopsy samples from patients, preventing the adjustment of treatment programmes in a longitudinal monitoring precision medicine approach. A requirement for larger sample volumes is also more resource intensive and increases experimental processing times. Comparatively, evidence suggests that liquid biopsy methods to detect cfDNA typically require a <5 mL starting volume [100]. However, the aforementioned survey failed to factor whether samples were derived from plasma, serum, or urine preventing effective comparison between enrichment methodologies. Additionally, Gardiner et al. (2016) recorded feedback from multiple individuals within the same department, potentially introducing selection bias due to the high likelihood of these investigators utilising similar equipment and isolation methods such as those detailed in Figure 3 [99].

A seminal paper by Stam et al. (2021) provides a comprehensive overview of various EV isolation techniques, indicating that combined enrichment methods account for ~60% of current EV isolations [101,102]. In light of this methodology appraisal, the most suitable technique for an EV-based diagnostic assay is suggested to be immunocapture, owing to its high purity, resolution, and ability to be combined with light scattering flow cytometry for the accurate assessment of EV subpopulations [101]. However, limited attention has been given to the scalability and potential commercialisation of this method, which is hindered by the use of expensive antibodies with inherent limitations in shelf life and production complexity [103]. Evidence also indicates a necessity for sample pre-processing before EV capture, as immunocapture cannot effectively isolate EVs from complex matrices such as blood plasma or serum [104]. This inefficiency is attributed to the presence of numerous competing antibody binding sites within complex samples, leading to reduced yield and EV purity. Overall, the EV-diagnostics field indicates a need to shift to standardised and more suitable/widely accessible technologies applicable to scale up commercialisation. Literature further underscores substantial variability in EV enumeration and characterisation, indicating potential adverse implications for downstream clinical decision-making [89,105,106,107]. Ultimately, this diminishes the likelihood of EV-based biomarkers successfully navigating the essential stages of analytical/clinical validation and utility endorsement, potentially contributing to the limited numbers of ongoing clinical trials.

## 4. Conclusions and Future Directions

In summary, despite some contradiction, tdEVs have emerged as a promising new class of cancer biomarkers, due to their unique capacity to harbour cancer-derived cargo reflective of their cell-of-origin. Their detection through liquid biopsy holds potential for offering crucial diagnostic and prognostic insights before, during, and after patient treatment ultimately assisting clinicians in optimising patient care. However, the field of EVs is new, and there is a current lack of understanding of their functional role within cancer pathogenesis. Therefore, the full extent of EVs’ potential in cancer diagnosis and treatment remains unclear. Although the majority of literature expresses a favourable view on the potential utilisation of EVs, particularly with interest in harnessing EVs as drug delivery platforms for cancer therapy, few acknowledge or discuss the considerable technological advancements needed to overcome existing limitations [108,109]. For example, the current lack of standardised methodologies for tdEV enrichment from host EV populations causes inconsistencies in the subpopulations isolated, hindering their characterisation and validation, and thus represents the largest obstacle facing the development of novel EV-based biomarkers. This indicates a clear need for the diagnostic field to develop significantly more reliable EV isolation techniques/methods, and for governing bodies to standardise such extraction methods and analytical platforms. This will increase the prospects of an EV-based cancer biomarker passing the analytical and clinical validation framework set by international regulatory agencies.

As EVs inherently represent the constituent cells from which they were released, they provide a snapshot of the heterogeneity present within tumours. This heterogeneity may pose a significant challenge to overcome to detect subtle differences in tdEV cargo, interpret their clinical significance amidst diverse cellular profiles, and to translate findings into effective diagnostic and therapeutic strategies. The research field stands to gain from directing efforts towards the development of economically feasible and widely accessible techniques for enriching tdEVs, both of which are key considerations for the clinical translation and widespread adoption of a diagnostic biomarker. An interesting avenue to explore for the future of immunocapture could involve leveraging multiple distinct tdEV surface markers through innovative bi-specific antibodies, reflecting the contemporary next-generation approach to antibody-drug conjugates [110,111]. Using one bi-specific antibody to simultaneously target two different tumour-derived epitopes holds promise for increasing assay specificity while facilitating the capture of multiple tdEV subpopulations in a single step. This approach circumvents the requirement for the development of multiple monoclonal antibodies each specific for a single tdEV population, thereby streamlining assay design and implementation.

Overall, there is a clear need to advance our understanding of the emerging role of EVs in the context of cancer communication and pathogenesis, to identify their biological significance, and to establish robust associations with clinical phenotypes. Concerted efforts are required to develop and refine technologies that offer improved accuracy and reliability in assays to unlock the translational potential of EV-based cancer biomarkers for commercial use.

## Figures and Tables

**Figure 1 biology-13-00694-f001:**
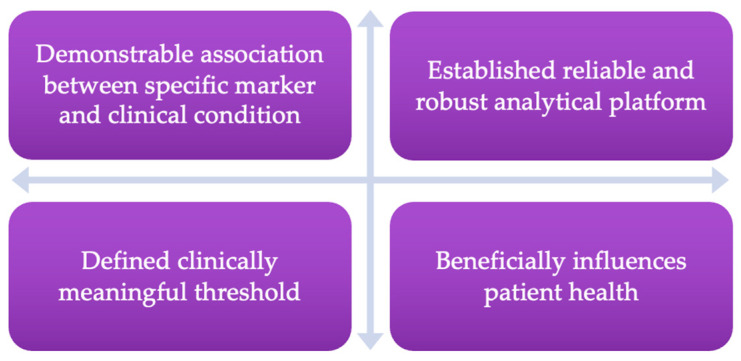
Important characteristics of a clinical biomarker.

**Figure 2 biology-13-00694-f002:**
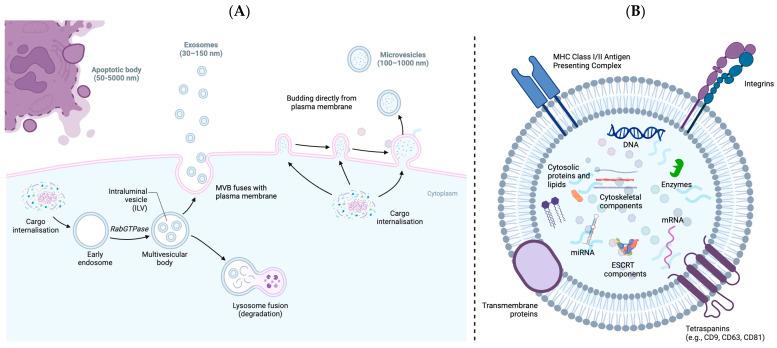
(**A**) Illustration of EV biogenesis pathways depicting the secretion of exosomes (30–100 nm) through multivesicular bodies within endosomes, budding of the plasma membrane generating microvesicles (100–1000 nm), and apoptotic bodies (50–5000 nm), which are released by membrane budding as a byproduct of programmed cell death. (**B**) A typical EV will contain a variety of macromolecules including coding and non-coding species of RNA, DNA, signalling proteins, lipids, and transcriptional regulators. ESCRT, endosomal sorting complexes required for transport; MHC, major histocompatibility complex.

**Figure 3 biology-13-00694-f003:**
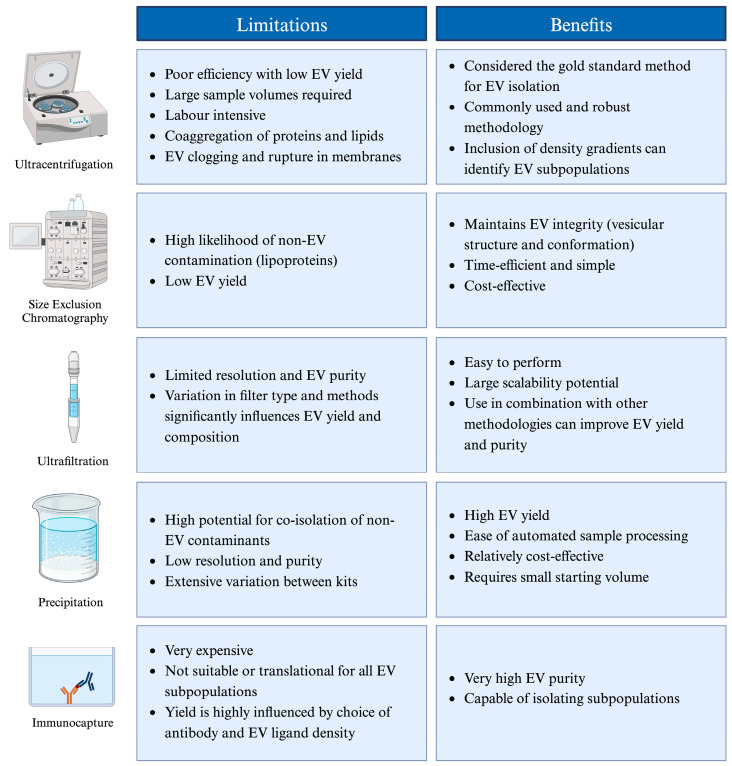
A summary of the limitations and benefits of classically used experimental methodologies used for EV isolation and enrichment.

**Table 3 biology-13-00694-t003:** Examples of common EV enrichment contaminants.

Contaminant	Subtype
Lipoproteins	High-density
	Low-density
	Very low-density
	Chylomicrons
Proteins	Aggregates
	Soluble proteins
	Ribonucleoproteins
Nucleic acids	Circulating free RNA/DNA
Other	Cellular debris
	Apoptotic bodies
	Viruses

## Data Availability

Not applicable.

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
