# Peer review of "Prospects and Current Challenges of Extracellular Vesicle-Based Biomarkers in Cancer"

_biology, 2024, doi:10.3390/biology13090694_

Round 1

Reviewer 1 Report

Comments and Suggestions for Authors

The authors summarized the importance and difficulties of EVs as biomarkers for early detection of cancer. This topic will be of interest to scientists in the field of cancer research from basic to translational science. The current version of the manuscript is well-written with updated information and is suitable for publication. I only have some minor comments below.

Line 59: Please correct “a clinically” to “a clinical”.

Line 92: Please add “U.S.” before “National Lung…”.

Line 124: Biofluids can include stool which may contain EVs and blood. Indeed, the stool test kit for colon cancer screening is currently available clinically, which is not discussed in this review.

Lines 138, 141 and 209: There is an extra bracket after the reference number.

Line 168: Please add space after “times”.

Line 299: I would add the stool screening information in Table 1. However, I would like to leave it to the authors to decide whether to add or not.

Line 358: This is a well-organized summary of the limitations and benefits of EV analysis, which needs a Figure number and Figure title so that the authors and readers can easily refer.

Author Response

Reviewer 1:

The authors summarized the importance and difficulties of EVs as biomarkers for early detection of cancer. This topic will be of interest to scientists in the field of cancer research from basic to translational science. The current version of the manuscript is well-written with updated information and is suitable for publication. I only have some minor comments below.

1) Line 59: Please correct “a clinically” to “a clinical”.

Response: Thank you for identifying our typos, we have now amended the wording in the Figure 1 legend.

2) Line 92: Please add “U.S.” before “National Lung…”.

Response:  We agree with this comment and have included U.S. in line 100

3) Line 124: Biofluids can include stool which may contain EVs and blood. Indeed, the stool test kit for colon cancer screening is currently available clinically, which is not discussed in this review.

Response:  We thank the reviewer for this comment and have now added stool to the list of biofluids from which EVs can be assessed in line 133.

4) Lines 138, 141 and 209: There is an extra bracket after the reference number.

Response: Thank you for identifying these typos, we have now corrected this in the revised version.

5) Line 168: Please add space after “times”.

Response: Thank you for identifying these typos, we have now corrected this in the revised version.

6) Line 299: I would add the stool screening information in Table 1. However, I would like to leave it to the authors to decide whether to add or not.

Response: We thank the reviewer for their suggestion, and we have now amended the table (Table 4) to include the Faecal Immunochemical Test as a clinically validated biomarker test for colorectal cancer.

7) Line 358: This is a well-organized summary of the limitations and benefits of EV analysis, which needs a Figure number and Figure title so that the authors and readers can easily refer.

Response: We thank the reviewer for this suggestion and apologise for this oversight. We have now added a Figure number and legend in the revised version.

Reviewer 2 Report

Comments and Suggestions for Authors

This is a timely and complete review of the potential for EVs as clinical biomarkers. It is well written and covers the field in a useful manner. Things that would improve the review include:

1. A table or figure summarizing the different types of EVs and their approximate sizes. 'Large' or 'small' EVs are referred to throughout the manuscript and this should be specified what is meant by this.

2. optimized dynamic light scattering is presented as a viable and important potential clinical option for EV detection in the text, yet it is not included in Figure 2 (and there is no figure legend for figure 2).

3. Line 23 "size, density, biochemical composition and cell of origin" - this could use some follow up specifics; what sizes? what type of biochemical compositions? what cell types are different?

4. lines 231 -237: a summary table of commonly found markers and the EV associated with them would be useful here

5. section 3.2: a list of common contaminants would be helpful

6. The following sentence: "Serum starvation during...biology experiments" line 265: It is unclear what this means. Serum starvation is done for many reasons: to pulse/chase cellular events, to induce quiescence, etc. Please be more specific.

Author Response

This is a timely and complete review of the potential for EVs as clinical biomarkers. It is well written and covers the field in a useful manner. Things that would improve the review include:

1) A table or figure summarizing the different types of EVs and their approximate sizes. 'Large' or 'small' EVs are referred to throughout the manuscript and this should be specified what is meant by this.

Response: We thank the reviewer for this suggestion. We have now amended the manuscript to include a new figure (Figure 2) which illustrates simplified EV biogenesis pathways into exosomes, microvesicles, and apoptotic bodies which represent the largest categories of EVs, along with details of their sizes.

2) Optimized dynamic light scattering is presented as a viable and important potential clinical option for EV detection in the text, yet it is not included in Figure 2 (and there is no figure legend for figure 2).

Response: We thank the reviewer for their comment. As mentioned, DLS does offer an important avenue for detection of pathological EVs based on their size and shape. However, Figure 2 discusses routinely used experimental techniques for isolation and enrichment of EVs and therefore we believe that DLS does not merit inclusion into that figure.

We also apologise for the oversight in including a figure number and legend and we have now corrected this in the revised manuscript.

3) Line 223 "size, density, biochemical composition and cell of origin" - this could use some follow up specifics; what sizes? what type of biochemical compositions? what cell types are different?

Response: We thank the reviewer for their comment. We have now added a new figure (Figure 2) illustrating the biogenesis pathways and different sizes of resultant EVs, and the typical composition of EVs. We have also added a new table (Table 2) to highlight markers for the identification of specific EV populations and cell-of-origin. We have also added the following text to the revised manuscript at line 252:

‘Traditional classification of EVs condenses the wide range of particle types into three categories: exosomes, which are formed through the exocytosis of multivesicular endo-somes (30-150 nm diameter, 1.13-1.19 g/mL); microvesicles, which are generated by outward blebbing of the plasma membrane (100-1,000 nm diameter, 1.08–1.19 g/mL); apoptotic bodies, which are remnants of the late stages of apoptosis (50-5,000 nm diameter, 1.12–1.23 g/mL) (Figure 2). Notably, the biochemical composition of EVs is highly diverse and reflects both common EV markers and cell-specific components (see Table 2).’

4) lines 231 -237: a summary table of commonly found markers and the EV associated with them would be useful here.

Response: We thank the reviewer for their suggestion. We have now added a new table (Table 2) summarising the commonly found markers.

5) section 3.2: a list of common contaminants would be helpful.

Response: We thank the reviewer for their suggestion. We have now added a new table (Table 3) summarising the common contaminants of EV isolation.

6) The following sentence: "Serum starvation during...biology experiments" line 265: It is unclear what this means. Serum starvation is done for many reasons: to pulse/chase cellular events, to induce quiescence, etc. Please be more specific.

Response: We thank the reviewer for their comment. We have now added the following text as clarification at line 315:

‘Serum starvation during in vitro cell culture research is a versatile tool for experimental manipulation, commonly employed for purposes involving cell cycle synchronisation, studying cellular metabolic stress responses by serving as a model for nutrient deprivation, and manipulating cell differentiation and gene expression programmes [61].’

Reviewer 3 Report

Comments and Suggestions for Authors

In this article S.R. Lawrence & K.M. Shah describe the application of extracellular vesicles (EVs) as cancer biomarkers, highlighting its potential as well as current challenges hindering its clinical translation.

Overall the authors concisely pointed out the prospects of EVs over other liquid biomarkers while comprehensively explaining limitations faced, regarding its isolation processes and downstream analyses. 

The length of the review is very proper and I believe the article warrants great interests from the readers of Biology. I would suggest the acceptance of this article following several minor comments below:

1) Schematic illustration highlighting EV features and contents will be very helpful for the understanding of broader audiences.

2) Highly recommend the authors to put one summarizing table in the discussion part to highlight the challenges mentioned, examples from references, and potential solutions to address such issue.

3) Related to future prospects, strongly encourage the authors to discuss which challenge is most detrimental for the eventual clinical translation of EV-based diagnosis.

Author Response

In this article S.R. Lawrence & K.M. Shah describe the application of extracellular vesicles (EVs) as cancer biomarkers, highlighting its potential as well as current challenges hindering its clinical translation. Overall, the authors concisely pointed out the prospects of EVs over other liquid biomarkers while comprehensively explaining limitations faced, regarding its isolation processes and downstream analyses. The length of the review is very proper and I believe the article warrants great interests from the readers of Biology. I would suggest the acceptance of this article following several minor comments below:

1) Schematic illustration highlighting EV features and contents will be very helpful for the understanding of broader audiences.

Response: We thank the reviewer for their suggestion. We have now added a new Figure (Figure 2) that illustrates the biogenesis pathways of EVs and the common characteristics and contents of EVs.

2) Highly recommend the authors to put one summarizing table in the discussion part to highlight the challenges mentioned, examples from references, and potential solutions to address such issue.

Response: We thank the reviewer for their suggestion. We have now added a new Table (Table 1) that summarises the main challenges and potential solutions to enable clinical translation of tdEV as biomarkers from the discussions in the main text.

3) Related to future prospects, strongly encourage the authors to discuss which challenge is most detrimental for the eventual clinical translation of EV-based diagnosis.

Response: We thank the reviewer for their suggestion. In our opinion the most important barrier to clinical translation of EV-based diagnosis is the lack of standardisation in isolation and characterisation of EV populations from biofluids. We have now added the following text at lines 421-428

‘the current lack of standardised methodologies for tdEV enrichment from host EV populations causes inconsistencies in the subpopulations isolated, hindering their characterisation and validation, and thus represents the largest obstacle facing the development of novel EV-based biomarkers. This indicates a clear need for the diagnostic field to develop significantly more reliable EV isolation techniques/methods, and for governing bodies to standardise such extraction methods and analytical platforms. This will increase the prospects of an EV-based cancer biomarker passing the analytical and clinical validation framework as set by international regulatory agencies.’

Reviewer 4 Report

Comments and Suggestions for Authors

Comments- 3164980

  1. In the review paper titled “Prospects and Current Challenges of Extracellular Vesicle-Based Biomarkers in Cancer,” the author effectively discussed the potential and challenges of tumour- derived extracellular vesicles (tdEVs) in liquid biopsy for better diagnosis, and control in cancer patients.
  2. The review is well-crafted with a strong correlation between tdEVs and cancer prognosis, along with insightful diagnostic methods to reduce economic burden in cancer treatment.
  3. The most interesting part of the review is that discussion on analysis of EVs using dynamic light-scattering. However, it would be more interesting if the author had discussed in detail about atomic spectroscopy advantages and challenges in detecting EVs. It must be added.
  4. In the introduction, the author discussed the components of EVs, such as RNA/DNA, which are distinct from the host EV population. It would be clear if the author had mentioned the difference, whether tdEVs nucleic acids contain any specific modifications or mutations. It must be added.
  5. In the prospects section, the author mentioned the detection of EVs from biofluids such as urine and saliva, but there are no more detailed detection methods than real-time PCR. It would be useful if there are any other detection methods for urine and saliva for detecting altered proteins. It must be added.
  6. The author did not discuss ELISA, which is also one of the reliable methods to detect tdEVs, is there any specific reason that ELISA was not included and discussed? It must be added.
  7. In Table 1, the author has mentioned sample type as blood for circulating tumor cells and circulating cfDNA; it would be better if the author had mentioned plasma or serum. It must be added.
  8. The author should include the pictorial representation of extracellular vesicles such as exosomes, apoptotic bodies, etc., and how they are delivered, targeted, or interact with neighboring cells in the introduction, as it’s the main theme of the review. It would be clearer for the readers. It must be added.
  9. The figure explaining the limitations and benefits of EV extraction and processing methods should need a figure number and legends.
  10. Overall, the review provides a detailed and worthful analysis of the current challenges of EV-based cancer diagnostics. The author effectively discussed the EV heterogeneity and contamination and storage issues.

I recommend this article for acceptance after addressing all the above questions by the author.

Author Response

Reviewer 4:

1) In the review paper titled “Prospects and Current Challenges of Extracellular Vesicle-Based Biomarkers in Cancer,” the author effectively discussed the potential and challenges of tumour- derived extracellular vesicles (tdEVs) in liquid biopsy for better diagnosis, and control in cancer patients.

Response: We thank the reviewer for their comment.

2) The review is well-crafted with a strong correlation between tdEVs and cancer prognosis, along with insightful diagnostic methods to reduce economic burden in cancer treatment.

Response: We thank the reviewer and are pleased that they believe that the review is well-crafted.

3) The most interesting part of the review is that discussion on analysis of EVs using dynamic light-scattering. However, it would be more interesting if the author had discussed in detail about atomic spectroscopy advantages and challenges in detecting EVs. It must be added.

Response: We thank the reviewer for their suggestion. We agree that elaboration on atomic force microscopy would prove beneficial in highlighting a novel technique in EV imaging. We have added a description on how AFM operates, the advantages of using this methodology in the biophysical characterisation of EVs including evidence supporting its role in generating novel insights into tdEV properties, and some limitations of the workflow. Specifically, we have added the following text at lines 186-205:

‘This type of nanotechnology is capable of atomic scale imaging operating through a cantilever equipped with a nanoscale tip, which can scan the surface of an EV using a defined tip force while a laser beam reflected onto a photodiode precisely monitors deflection of the cantilever, thus generating topographic images of the EV’s surface [34]. An advantage of using AFM for the biophysical characterisation of EVs can be attributed to the technology’s capability to deposit analyte on a solid substrate and image samples either dried or within cell culture media, with no need for staining or fixation providing a degree of flexibility within the workflow [35]. The ability to analyse and generate topographic images of single membrane proteins at a lateral and vertical resolution of <1 nm and <0.1 nm respectively, while also generating insights into specific vesicular mechanical characteristics suggests clinical potential for the identification of tdEV nanomechanical fingerprints. One study using AFM to investigate mechanical properties of human breast cancer-derived tdEVs identified that vesicle stiffness and osmotic pressure increased with in-creasing disease malignancy, indicating that this technology may also aid in patient stratification into low and high-risk groups [36]. AFM’s utility is however limited to surface analysis and low throughput particle engagement, and can only analyse several hundred nanosized particles within an hour [37]. The workflow still requires significant sample processing and is limited by the bottleneck of pure EV enrichment, suggesting that AFM can only aid in the identification of specific tdEVs and provide an indication of contaminant levels when used in conjunction with other diagnostic techniques.’

4) In the introduction, the author discussed the components of EVs, such as RNA/DNA, which are distinct from the host EV population. It would be clear if the author had mentioned the difference, whether tdEVs nucleic acids contain any specific modifications or mutations. It must be added.

Response: We thank the reviewer for their comment and suggestion. We agree with this comment that it would be beneficial to provide an example of how tdEV DNA might reflect the mutational landscape of the cells of origin. We have now discussed a study demonstrating the presence of EVs harbouring mutations in numerous melanoma-related genes that mirror the originating tumour. Lines 75-82.

‘For instance, tdEVs have been found to carry DNA that reflects the mutational landscape of their cells of origin. Within the context of melanoma, one study isolated EVs from six human melanoma metastatic tissues and compared the DNA mutations with both tumour tissue DNA and plasma DNA. Using ultrasensitive sequencing (SiMSen-seq), researchers investigated a panel of 34 melanoma-related genes. Notably, mutations were detected in six genes (BRAF, NRAS, CDKN2A, STK19, PPP6C) in the EVs isolated from melanoma tissues, closely mirroring the mutational profile of the originating tumours [12]’

5) In the prospects section, the author mentioned the detection of EVs from biofluids such as urine and saliva, but there are no more detailed detection methods than real-time PCR. It would be useful if there are any other detection methods for urine and saliva for detecting altered proteins. It must be added.

Response:  We thank the reviewer for their comment. In addition to real-time PCR, we had mentioned the use of dynamic light scattering (DLS) (line 167). We have now included other detection strategies including atomic force microscopy (AFM) (line 185), as well as Mass Spectrometry (MS) and ELISAs (line 298-306; please see the comment below).

6) The author did not discuss ELISA, which is also one of the reliable methods to detect tdEVs, is there any specific reason that ELISA was not included and discussed? It must be added.

Response: We thank the reviewer for this suggestion. We agree with this comment that it would be good to mention some other methodologies that are used for the detection of altered proteins. This amendment can be identified at Line 298-306 as below:

‘This was completed using high resolution/high mass accuracy nano-liquid chromatography tandem Mass Spectrometry (MS) data with validation and quantitation of proteins performed experimentally using Enzyme-Linked Immunosorbent Assays (ELISAs). These findings suggest the prospective ability of tdEV proteomic analysis to discriminate be-tween cancer and non-cancer patients, highlighting the importance of further investigation in this area [69]. Following EV enrichment, commonly used methods to interrogate the proteomic landscape include Western blotting, ELISAs, MS, and flow cytometry which are powerful tools to detect and quantify tdEV-specific proteins. However, despite being highly sensitive techniques they appear impractical for clinical use due to their labour/resource intensive nature [70].’

7) In Table 1, the author has mentioned sample type as blood for circulating tumor cells and circulating cfDNA; it would be better if the author had mentioned plasma or serum. It must be added.

Response: We agree with this observation, and we have now added these details in the Table (now Table 4).

8) The author should include the pictorial representation of extracellular vesicles such as exosomes, apoptotic bodies, etc., and how they are delivered, targeted, or interact with neighbouring cells in the introduction, as it’s the main theme of the review. It would be clearer for the readers. It must be added.

Response: We thank the reviewer for this suggestion. We have added a new figure (Figure 2) illustrating a simplified EV biogenesis pathways into exosomes, microvesicles, and apoptotic bodies which represent the largest categories of EVs. We have also included a representative image depicting a typical EVs contents.

9) The figure explaining the limitations and benefits of EV extraction and processing methods should need a figure number and legends.

Response: We also apologise for the oversight in including a figure number and legend and we have now corrected this in the revised manuscript.

10) Overall, the review provides a detailed and worthful analysis of the current challenges of EV-based cancer diagnostics. The author effectively discussed the EV heterogeneity and contamination and storage issues. I recommend this article for acceptance after addressing all the above questions by the author.

Response: We thank the reviewer for their comment and their recommendation.

Round 2

Reviewer 4 Report

Comments and Suggestions for Authors

Comments- 3164980

  1. In the review paper titled “Prospects and Current Challenges of Extracellular Vesicle-Based Biomarkers in Cancer,” the author effectively discussed the potential and challenges of tumour- derived extracellular vesicles (tdEVs) in liquid biopsy for better diagnosis, and control in cancer patients.
  2. In the first round of review, the author addressed all the comments carefully and answers are satisfactory, except the below comment.
  3. For the comment 4: In the introduction, the author discussed the components of EVs, such as RNA/DNA, which are distinct from the host EV population. It would be clear if the author had mentioned the difference, whether tdEVs nucleic acids contain any specific modifications or mutations. 

The author had addressed the comment partially; however, it would enhance clarity to include a discussion on the status of methylation or acetylation of tdEVs nucleic acids compared to normal nucleic acids. For example, DNA hypermethylation are important epigenetic biomarkers for cancer diagnosis. It must be added.

4.     I recommend this article for acceptance with minor revision after addressing the    above comment by the author.

Author Response

1. In the review paper titled “Prospects and Current Challenges of Extracellular Vesicle-Based Biomarkers in Cancer,” the author effectively discussed the potential and challenges of tumour- derived extracellular vesicles (tdEVs) in liquid biopsy for better diagnosis, and control in cancer patients. In the first round of review, the author addressed all the comments carefully and answers are satisfactory, except the below comment.

2. Comment:

For the comment 4: In the introduction, the author discussed the components of EVs, such as RNA/DNA, which are distinct from the host EV population. It would be clear if the author had mentioned the difference, whether tdEVs nucleic acids contain any specific modifications or mutations. 

The author had addressed the comment partially; however, it would enhance clarity to include a discussion on the status of methylation or acetylation of tdEVs nucleic acids compared to normal nucleic acids. For example, DNA hypermethylation are important epigenetic biomarkers for cancer diagnosis. It must be added.

Response:  We thank the reviewer for their suggestion. We agree that DNA methylation and histone acetylation of are important biomarkers for cancer diagnosis, and we have added the following paragraph at line 82 as per your suggestion:

‘It is known that in cancer development, hypermethylation of the DNA is an early event and it remains methylated as cancers progress [13]. Indeed, methylation and other epigenetic status can also be exploited as a diagnostic biomarker [14,15]. For instance, in gastric cancers, the methylation status of BarH-like homeobox protein (BARHL2) from gastric juice derived EVs has been shown to have 90% sensitivity and 100% specificity with respect to discriminating patients from controls [16]. More recently, Maire et al. (2021) demonstrated that the genome-wide methylation profile and copy number variations (CNV) of glioblastoma derived EVs mirror the landscape observed in the original tumours [17].

Histone acetylation has also shown to play an important role in tumorigenesis, and acetylome analyses of tdEVs from breast cancer cells have found a distinct acetylated protein profile that may mirror the cell of origin and thus serve as good biomarker candidates [18].’

The main focus of our review is to discuss the challenges for clinical translation of tdEVs as biomarkers, and there are more comprehensive reviews that focus on the emerging roles of extracellular vesicle DNA available which we have cited in our manuscript (PMIDs:  36900248, 26582468).

3. I recommend this article for acceptance with minor revision after addressing the    above comment by the author.

Response: Many thanks for your recommendation.